# Spatial Accessibility Analysis of Medical Facilities Based on Public Transportation Networks

**DOI:** 10.3390/ijerph192316224

**Published:** 2022-12-04

**Authors:** Ying Liu, Han Gu, Yuyu Shi

**Affiliations:** College of Architecture and Urban Planning, Chongqing Jiaotong University, Chongqing 400075, China

**Keywords:** spatial accessibility, medical facilities, multiple transportations, service area method, network analysis

## Abstract

Aiming to look at the problems of the unreasonable layout of medical facilities and low coverage of primary medical services. This paper selects tertiary grade A hospitals, general hospitals, specialized hospitals, community-level hospitals, clinics, and pharmacies in the main urban areas of Chongqing as research objects. The nearest analysis, kernel density, mean center, and standard deviational ellipse method were used to analyze the spatial differentiation characteristics of medical facilities and public transportation stations. Spatial accessibility was assessed from the perspective of service area ratios and service population ratios by constructing multiple modes of transportation (pedestrian systems, bus lines, rail lines). The results show that (1) the spatial layout of medical facilities in the main urban area of Chongqing is unbalanced; and the spatial distribution of medical facilities is characterized by “large agglomeration, small dispersion” and “multi-center group”; (2) the sub-core circle is centered on the Southwest University Area in Beibei District, the University Town Area in Shapingba, the Yudong Area and Lijiatuo Area in Banan District, the Pingan Light Rail Station Area in Dadukou District, the Chongqing No. 8 Middle School Area in Jiulongpo District, the Tea Garden Area in Nanan District, and the Jiangbei Airport Area in Yubei District; (3) the medical facilities with the weakest average accessibility are tertiary grade A hospitals, and the strongest are pharmacies; (4) the areas with vital average accessibility are Yuzhong District, Shapingba District, Dadukou District, and Nanan District.

## 1. Introduction

Health services are an integral part of a city’s public service system. The development of medical services plays a vital role in not only the development of sustainable cities, but also their spatial distribution directly affects the physical health of urban residents [1,2]. With the acceleration of the urbanization process, the scope of cities continues to expand, the urban infrastructure extends to the periphery of the town, and people’s living space continues to spread to the surrounding areas. Some unreasonable problems in the layout of medical resources have emerged. In recent years, the accessibility of measurement space has received more and more attention. As the most commonly used factor for assessing the spatial equity of public service facilities, accessibility reflects the supply and demand and spatial distribution of facilities along a road network [3,4]. Accessibility is expressed as the ease of access for residents from one point in space to another and reflects the ease of access to public health services. The accessibility intensity of medical facilities is related to whether it is convenient for urban residents to seek medical treatment and whether their health is guaranteed. Michael B. Teitz and Richard L. Morrill conducted an in-depth study of equity efficiency in utility location selection [5,6]. In 1959, the concept of accessibility was proposed by Hansen [7].

The rapid growth of the number of private cars has worsened the urban traffic situation, and the travel methods of urban residents have become more geared towards advocating public transportation that saves energy, reduces pollution, benefits health, and takes into account efficiency (Data from Chongqing Municipal Government (www.cq.gov.cn, accessed on 18 November 2022). In a city, buses and metros are the primary means of transportation [8]. According to the 2019 Chongqing Central Urban Transportation Development Report, public transportation has become the first choice for residents. In addition, in non-emergency situations, it is usual and reasonable for residents to reach medical facilities by public transportation. Therefore, it is even more practical to assess the accessibility of medical facilities via public transport. In the current research on the accessibility of public transportation, scholars have established road network datasets using urban pedestrian roads [9], buses [10,11], rail transit [12,13], and bicycles [14]. Most scholars compare and analyze the accessibility results under different modes of travel [15]. These studies focus on single-layer networks [16]. Only a few scholars combine multiple modes of transportation. Mao and Langford et al. assess spatial accessibility through various modes of transport, providing more accurate spatial accessibility measurements in multimodal environments [17,18]. This study combines the walking system with bus lines and rail lines and uses stations as connection points to build multiple transportation networks. This method restores the actual travel path (the transition from the nearest walking road to the nearest station rather than directly from the nearest station under the search threshold). The increase in the cost of walking time makes the experimental results more scientific.

The methods to measure spatial accessibility mainly include the population-to-provider ratio, the kernel density method [19], the shortest path method [20,21] the service area method, the potential model [17], and the two-step floating catchment area method (2SFCA) [1,19,22,23,24,25,26]. However, these methods have some drawbacks. The population-to-provider ratio is the ratio of medical capacity to the population of an administrative area, which is simple to calculate but ignores many factors, such as distance. The kernel density method uses GIS to plot the probability density function surface of the supply and demand side and then calculate the ratio of the two faces. One of the shortcomings of this method is that it does not consider the interaction between supply and demand. The shortest path method is considered suitable for assessing medical accessibility in rural areas. However, it is not valid in high-density population areas [27]. One of the drawbacks of the potential model is that the calculations are complex and require a lot of information, and the introduction of impedance coefficients produces many uncertainties. Furthermore, it does not account for the fact that accessibility generally has solid socioeconomic characteristics [28]. 2SFCA considers the impact of supply ratios on accessibility. However, supply ratios have certain limitations, including the exclusion of patient transit on small geographical scales [29], the effect of distance attenuation [30], and the lack of service threshold measurements [31,32]. Among them, the setting of service thresholds is a crucial part of the study, and the threshold radius has been determined by referring to the distance to the nearest provider or based on the number of people available to serve [33], the number of medical devices, and the number of medical personnel [13,34]. This method requires high information collection at medical facilities.

Compared with other accessibility methods, the service area method is highly interpretable, simple to calculate, and can more directly represent the accessibility differences in different regions. The service area method can be divided into time accessibility assessment and spatial accessibility assessment methods according to different threshold units. Time accessibility is expressed as a starting point of the station, based on the road network, to calculate the area or population that can be covered at a particular time. Compared with the spatial accessibility assessment method, the former overcomes the errors caused by the isolation of rivers and roads to a certain extent, improves accuracy, and retains the advantages of simple calculation methods and concise results. Most scholars choose the service area method for urban green space accessibility measurement, and few studies analyze medical accessibility [35,36]. The service area method under time constraints can well explain the social opportunities obtained by residents through travel at a specific threshold time [37]. However, there are still limitations in the time accessibility assessment method, such as calculating the time cost directly from the station, not considering stop time, etc., which will underestimate the travel time cost and make it different from the actual situation. Scholars integrate detailed travel information into their analyses [38,39]. Djurhuus et al. divide travel time into the following parts: in-vehicle, interchange, transfer, waiting, and egress time [40]. However, impedance settings for road network data have not been widely used. Therefore, when constructing the network model, the study combines urban roads with bus lines and rail lines, takes the station as the transfer connection point, and sets the stop time as the delay impedance time, which restores the time cost of actual travel.

According to the classification of medical resources in China Health, urban medical service institutions adopt a two-level service system of “hospital-community health service institutions” [41]. According to the different contents of hospital diagnosis and treatment, hospitals can be divided into general hospitals, traditional Chinese medicine hospitals (integrated traditional Chinese and Western medicine hospitals), specialized hospitals, etc. Community health service organizations provide preventive, medical, healthcare, rehabilitation, and other services to residents, such as clinics, community-level hospitals, nursing homes, etc. Currently, the number of spatial accessibility studies of community-level medical services in China is still limited. Most studies conducted equity studies on accessibility at different levels of healthcare facilities, depending on the scope of hospital services [42,43,44]. The benefits of primary healthcare facilities such as community-level hospitals, clinics, and pharmacies are overlooked [45,46,47,48,49]. Therefore, for more careful consideration, tertiary grade A hospitals (due to the particularity of the scope, this is considered a separate category), general hospitals, and specialized hospitals were selected as representatives of medical facilities. In addition, community-level hospitals, clinics, and pharmacies were selected as representatives of community health service organizations.

Most of the studies on the spatial accessibility of medical facilities are aimed at developed coastal cities, with less involving cities in the less developed western regions. Most of the studies choose cities in plain areas and rarely analyze the accessibility of medical services in mountain cities. Coupled with the complex terrain of mountainous areas and inconvenient transportation, the difference in medical accessibility will be tremendous and significantly different from that of plain cities. Chongqing officially became the second batch of pilot provinces and cities for comprehensive medical reform in 2017, and the configuration of medical facilities in Chongqing has gradually improved. However, there are still problems, such as unreasonable structural layout, a low level of public health services in some areas, and fragmentation of the service system. According to the data of the seventh census, the permanent population of Chongqing’s main urban area reached 10.34 million, an increase of 2.886 million in 10 years. For the Chongqing municipal government, providing adequate medical services for such a large population is challenging. In addition, in the current accessibility research on Chongqing, most scholars choose 2SFCA to analyze the accessibility of medical [50], food [51], transportation [16], etc. provision and have not used the service area method for analysis. Therefore, as a typical mountainous city, Chongqing has been chosen as a strong representative case for the study.

In general, based on geographic information system (GIS) network analysis and spatial analysis methods, this study constructs multiple transportation networks under three modes of travel: walking, rail transit, and bus. Evaluation of the fairness of medical facilities was based on the ratio of the area served and the population ratio. Specific research objectives were as follows:
(1)Compare the distribution characteristics of medical facilities and public transportation stations;(2)Explore the spatial differences in the accessibility of medical resources at all levels in the main urban area of Chongqing, and analyze the degree of balance between the regions;(3)Propose countermeasures and suggestions that can improve the spatial layout of local medical facilities. The analysis of the accessibility of medical facilities is conducive to the rational allocation of limited medical resources, promotes the equal structure of medical infrastructure, improves the service level of public health, and provides a reference for the medical facility layout research field.

The remainder of this paper is organized as follows: The study areas, associated data, and accessibility calculation methods are presented in detail in Section 2. Then, the spatial distribution and accessibility results of medical facilities are presented in Section 3. Finally, the conclusions and future work are presented in Section 4.

## 2. Materials and Methods

### 2.1. Study Area and Data Sources

Chongqing is located in the upper reaches of the Yangtze River. As a typical mountainous city, the main city of Chongqing has the spatial pattern of “two rivers and four veins”, so the main urban area of Chongqing is a typical polycentric city. The nine districts of Chongqing’s main urban area (Yuzhong District, Jiangbei District, Nanan District, Jiulongpo District, Shapingba District, Dadukou District, Beibei District, Yubei District, and Banan District) have a total area of 5467.21 square kilometers (Figure 1). As of 2020, the permanent population of the main urban area reached 10.3626 million, the population density reached 1894.79 people per km², and the urbanization rate reached 92.63%.

This study analyzes the accessibility of medical care in the major urban areas of Chongqing. All kinds of medicine in the main urban area of Chongqing are obtained from the points of interest (POI) of Baidu Maps. After classification, 13,754 medical facilities with spatial location data (including tertiary grade A hospitals, general hospitals, specialist hospitals, community-level hospitals, clinics, and pharmacies) have been obtained.

The administrative boundary vector data and population raster data for 2015 were obtained from the Resources and Environmental Sciences and Data Center. Urban road data (excluding the city ring, viaducts, highways and level 4 roads within the community), urban rail lines, bus routes, and station data were obtained from AutoNavi Maps. All data were updated as of April 2021, as shown in Figure 2.

### 2.2. Methods

#### 2.2.1. Spatial Accessibility Calculation

This study used the service area method to measure spatial accessibility at medical facilities. The accessibility of each facility is evaluated by comparing the service area and the number of people who can reach the medical facility within a specified time. The higher the proportion of population and area served under the time threshold, the higher the accessibility of medical facilities, as shown in Equations (1) and (2):SPR = sp/tp(1)
SAR = sa/ta(2)
where SPR represents the average population ratio, SAR represents the average service area ratio, sp represents the service area population, tp represents the total population, sa represents the service area, and ta represents the whole area.

In this paper, the network dataset was established by three modes of travel: walking, rail transit, and bus. Using the network analysis method, the service range under 15 min, 30 min, and 60 min is calculated from the medical facility as the starting point to obtain the spatial accessibility of the medical facility in each district. It is important to note that stations connect the three modes of transportation. The stop time was 30 s. The speed setting is based on the “Highway Engineering Technical Standard (JTGB01-2014)” and the “Urban Road Engineering Design Code”, combined with the influence of mountainous terrain on driving speed and related studies [50,52]. The walking speed was 1.2 m/s, the rail transit speed was 70 km/h, and the speed of buses was 40 km/h.

#### 2.2.2. Near Analysis

The near analysis was utilized to explore the spatial distribution of various medical points in the main urban areas of Chongqing. The near analysis is calculated based on the average distance between the feature and the most immediate feature so that it can more scientifically describe the degree of distribution of point-like things in geographical space, and its model is as follows:(3)$$R=\frac{\bar{r_{1}}}{\bar{r_{\varepsilon }}}=\bar{r_{1}}\times 2\sqrt{\frac{n}{A}}$$
where *R* represents the nearest neighbor ratio, $\bar{r_{1}}$ represents the average of the distances between the nearest points, $\bar{r_{\varepsilon }}$ represents the theoretical average distance, *n* represents the number of study subjects, and *A* means the area of the study area. *R* = 1 indicates a random distribution. If the value of *R* exceeds 1, this shows that the study object is evenly distributed. If the R value is less than 1, the subjects show a cohesive type, and a smaller value indicates a more substantial spatial aggregation.

#### 2.2.3. Kernel Density

To explore the spatial distribution of medical and public transport, this paper uses the kernel density method to analyze the unit area of medical points in the study area, which was implemented in ArcGIS 10.2 software.
(4)$$D=\frac{3{\left(1-scale^{2}\right)}^{2}}{\pi r^{2}}$$
where *r* represents the radius of the lookup, and the scale represents the ratio of the distance from the center of the raster to the point and line.

#### 2.2.4. Spatial Distribution Trends

In order to explore the spatial distribution centers and distribution trends of medical facilities in the main urban areas of Chongqing, the mean center and standard deviational ellipse methods were used to evaluate it, and both were implemented through ArcGIS 10.2 software. The mean center method obtains the geographic center of a medical facility by calculating the average of the *x* and *y* coordinates, as shown in Equations (5) and (6).
(5)$$\bar{Y}=\overset{N}{\underset{i=1}{\sum }}\frac{Y_{i}}{N}$$
(6)$$\bar{X}=\overset{N}{\underset{i=1}{\sum }}\frac{X_{i}}{N}$$

The standard deviational ellipse method represents the distribution trend in space, the XstdDist field reflects the direction of the spatial distribution, and the YstdDist field is the opposite, which is calculated as follows (7):(7)$$C=\left(\begin{array}{cc} var\left(x\right) & cov\left(x,y\right)\\ cov\left(y,x\right) & var\left(y\right) \end{array}\right)=\frac{1}{n}\left(\begin{array}{cc} \overset{n}{\underset{i=1}{\sum }}{\tilde{x}}_{i}^{2} & \overset{n}{\underset{i=1}{\sum }}\tilde{x_{i}}\tilde{y_{i}}\\ \overset{n}{\underset{i=1}{\sum }}\tilde{x_{i}}\tilde{y_{i}} & \overset{n}{\underset{i=1}{\sum }}{\tilde{y}}_{i}^{2} \end{array}\right)$$
where *x* and *y* represent the coordinates of *i* feature, {$\bar{x},\bar{y}$} represent the average center of the feature, and *n* represents the total number.

## 3. Results

### 3.1. Characteristics of the Spatial Pattern of Medical Facilities

Based on the near analysis, the spatial distribution of medical facilities was obtained (Table 1). A significant spatial agglomeration characteristic was found in the study area. Among them, community-level hospitals have the lowest concentration, the R value was 0.6, and the space tends to be evenly distributed. This was followed by the general hospital, with an R value of 0.41. Pharmacies have the highest concentration, and the R value was 0.21. It is worth noting that the degree of aggregation of the six medical facilities was quite different, which is related to the function of different types of medical facilities. The R values for rail stations and bus stops were 0.41 and 0.48, respectively (Table 2).

The average nearest neighbor ratio of public transportation stations and medical points was similar, and both show the characteristics of different degrees of aggregation.

Figure 3 and Figure 4 show the difference in the distribution trend of medical facilities and public transportation in the major urban areas of Chongqing. The spatial distribution of most medical facilities has a long axis, except for tertiary grade A hospitals, which is a southeast–northwest trend. The spatial distribution of public transport stations was consistent with most medical facilities. The spatial distribution centers were located in Yubei District, and the spatial distribution trend was also in a south–north direction.

The spatial distribution of medical facilities is characterized by “large agglomeration, small dispersion” and “multi-center group”. Yuzhong District is the core circle layer. The sub-core circle is centered on the Southwest University Area in Beibei District, the University Town Area in Shapingba, the Yudong Area and Lijiatuo Area in Banan District, the Pingan Light Rail Station Area in Dadukou District, the Chongqing No. 8 Middle School Area in Jiulongpo District, the Tea Garden Area in Nanan District, and the Jiangbei Airport Area in Yubei District.

### 3.2. Spatial Accessibility Analysis of Various Types of Medical Facilities

#### 3.2.1. Tertiary Grade A Hospitals’ Accessibility

Calculations from Equations (1) and (2) have shown that tertiary grade A hospitals have the lowest average accessibility among all types of medical facilities, mainly because tertiary grade A hospitals are primarily located in the old city. 82% of the residents of Yuzhong District can reach the nearest tertiary grade A hospital within 15 min for medical treatment. Most residents of Yuzhong District, Shapingba District, Dadukou District, and Nanan District (50%) can reach the nearest tertiary first-class hospital within 60 min (Figure 5).

#### 3.2.2. General Hospitals’ Accessibility

Under the travel time of 15 min, the areas with a high service area ratio of general hospitals are 99.98% in Yuzhong District, 46.05% in Dadukou District, and 45.07% in Nanan District, and 99.99% in Yuzhong District, 51.53% in Dadukou District and 44.42% in Nanan District with high service population ratios. The service population ratio of Banan District, Beibei District, Jiangbei District, and Yubei District was below the average of 37.13%. Of these, only 8% of the residents of Banan could reach the nearest general hospital within 15 min. Under the travel time of 60 min, the areas with a high service population ratio are Shapingba District, 71.74%, Dadukou District, 67.13%, and Jiulongpo District, 62.22%.

#### 3.2.3. Specialist Hospitals’ Accessibility

Under the travel time of 15 min, the areas with a high service area of specialized hospitals are 99.98% in Yuzhong District, 53.30% in Dadukou District, and 52.69% in Shapingba District. The service population ratio of Banan District, Beibei District, Jiangbei District, and Yubei District is below the average of 41.56%, of which only 8% of the residents of Banan District can reach the nearest specialist hospital within 15 min. Under the travel time of 60 min, the areas with a higher service population were 71.28% in Shapingba District and 61.29% in Jiulongpo District.

#### 3.2.4. Community-Level Hospitals’ Accessibility

The distribution of community-level hospitals is characterized by equality and dispersion. The data show that at least 47% of residents in Yuzhong District, Shapingba District, Dadukou District, and Nanan District can reach the nearest community-level hospitals within 15 min. However, residents cannot quickly reach the nearest community-level hospitals in areas with large rural areas such as Banan District, Yubei District, and Beibei District, primarily rural areas. Among them, a minimal number of residents of Banan District (7%) can get to the nearest community-level hospitals within 15 min.

#### 3.2.5. Clinics’ Accessibility

The average accessibility of a clinic is vital. Under the travel time of 15 min, the average population ratio of a clinic reaches 42.10%, which is 5.0% higher than the average. The areas with a higher service population ratio are Yuzhong District at 99.7%, Shapingba District at 56.12%, Dadukou District at 52.03%, Nanan District at 50.35%, and Jiulongpo District at 44.08%. Within 60 min, the areas with the highest population ratio were Yuzhong District at 99.8%, Dadukou District at 89.77%, Shapingba District at 74.65%, and Jiulongpo District at 74.27%.

#### 3.2.6. Pharmacies’ Accessibility

Under the travel time of 15 min, the average population ratio served by pharmacies is the highest reaching 45.76%, 8.3% higher than the average. Under the travel time of 60 min, the average service population ratio of pharmacies reached 60.61%. The areas with higher service population ratios were Yuzhong District, 99.9%, Shapingba District, 72.68%, Dadukou District, 71.21%, Jiulongpo District, 71.65%, and Jiangbei District, 70.75%, but the population ratio of Banan District was only 18.93% because there is a large rural area in the south of the Banan District. Hence, the level of basic medical facilities is poor.

Generally speaking, pharmacy medical facilities are closest to people’s daily lives. Smaller pharmacies are densely distributed and have a significant overall number, making them more accessible and most frequently exposed to daily medical facilities. Residents with common or chronic diseases are also more likely to visit the nearest medical facility.

### 3.3. The Results of the Overall Accessibility Assessment

The average service area ratio and the average service population ratio of the tertiary grade A hospitals were 13.23% and 24.83%, respectively. The average service area ratio of pharmacies was 28.08%, and the average service population ratio was 45.83%. Therefore, the medical facilities with the weakest average accessibility are tertiary grade A hospitals and the strongest are pharmacies. At 15 min travel time, the average reachability intensity ranking is shown in Figure 6. The difference in the ratio of service population in various medical facilities is significant, and the difference between the maximum and minimum values is 29.51%. With a travel time of 30 min, the gap between the average accessibility level of various medical facilities has gradually narrowed. However, tertiary A hospitals’ average service population ratio is still below the average. The average service population ratio of general hospitals, community health centers and specialized hospitals was similar, and the difference was not more than 3%. Specialist hospitals are more accessible in a short period. However, over time, the average accessibility intensity has become more and more similar, mainly because there were more specialized hospitals than the first two, and the spatial distribution was more concentrated.

The areas with high average accessibility are Yuzhong District, Shapingba District, Dadukou District, and Nanan District. Under the travel time of 15 min, the service area ratio reached 99.9% in Yuzhong District, 43.67% in Nanan District, 39.39% in Shapingba District, and 35.67% in Dadukou District, and the average service population ratio reached 98.9% in Yuzhong District, 45.65% in Shapingba District, 44.76% in Dadukou District, and 42.84% in Nanan District. This reflects the ease of transportation and abundance of medical facilities available in these areas so that most people can access medical services in a shorter period. In the travel time of 15 min to 60 min, the average service population ratio of Banan District, Beibei District, Yubei District, and Jiangbei District has been below the average (Figure 7).

## 4. Discussion and Conclusions

In this study, near analysis and kernel density analysis were used to analyze the spatial distribution of medical facilities and public transportation stations in the main urban area of Chongqing. On this basis, accurate public transport network data and the service area method were used to assess the accessibility of medical facilities over different time periods. Demographic data are used as a proxy for the social context to analyze the equality of accessibility in medical facilities. In constructing the public transportation road network, the pedestrian system was combined with bus lines and rail lines and the station was used as a connection point to restore the actual travel path. In addition, this study also selected primary healthcare facilities such as community-level hospitals, pharmacies, and clinics as research subjects, making up for the current lack of access to primary healthcare facilities.

The study found that the degree of aggregation of various medical facilities varies greatly, mainly related to the functions they carry. In order to ensure the daily medical service needs, community-level hospitals were distributed within multiple neighborhoods on the principle of fairness and to serve a small range of residents. Therefore, the spatial pattern of community-level hospitals tends to be more decentralized. Clinics, specialized hospitals, and pharmacies need to integrate medical resources through a centralized layout because of their low level and fewer services. Finally, they meet the medical needs of urban residents, so their distribution and aggregation are higher. In contrast, general hospitals have the characteristics of a complete range of departments and a more comprehensive range of medical service needs. In daily life, patients are more likely to choose general hospitals for medical treatment, so the layout of general hospitals takes equality into account, and their spatial structure is more scattered in various districts.

Moreover, the spatial distribution of medical facilities is characterized by “large agglomeration, small dispersion” and “multi-center group”. The spatial distribution trend of medical facilities and public transportation stations is south–north, and the spatial distribution center is located in Yubei District. The areas with high average accessibility are Yuzhong District, Shapingba District, Dadukou District, and Nanan District. Pharmacies are the medical facilities with the strongest average accessibility, most residents (42%) can reach the nearest pharmacy within 15 min, and tertiary A hospitals have the weakest average accessibility, with only 11% of residents arriving within 15 min. At 15–30 min travel time, the average accessibility intensity of tertiary A hospitals and general hospitals has been below the average.

In this study, the service area method under the time threshold was selected to measure the accessibility of medical facilities. The accessibility intensity of different regions and medical facilities reflects the regions’ medical facilities and traffic accessibility. This method also applies to facility siting in different regions. Multiple transportation networks under the three modes of walking, rail transit, and bus were constructed, and the accessibility measurement of 15/30/60 min was selected to take into account the characteristics of the long traffic time of the mountain city of Chongqing in the spatial pattern of “multi-center group”. In plains cities, the time threshold will be relatively short. The travel time of Beijing urban rail transit accounts for the largest proportion at 40–50 min [53]. Our findings reveal that the distribution of medical facilities is related to Chongqing’s topography, transportation convenience, and population density. Extensive medical facilities are mainly located around traffic arteries. At the same time, small clinics and pharmacies are primarily distributed in densely populated community streets. In contrast, less accessible medical facilities are primarily located in traffic-congested old towns and sparsely populated new urban areas. The problem of inequality in accessibility was influenced by the distribution of medical facilities and public transportation stations, which provided ideas for the location and layout of cities belonging to mountainous terrain. The mountainous regions of the United States (Arizona, Colorado, New Mexico, Nevada, and Utah) use rail transit systems to help meet transportation needs and as distance from transit stations increases, real estate values fall [54]. Hong Kong, a representative of China’s mountain cities, found that public transportation provided a much faster travel time to the parks from all housing estates. The average travel time of crowded public housing residents exceeded that of private residential residents [55]. It follows that vulnerable groups are more affected by access to public transport and medical facilities.

Fairness can be horizontal or vertical [56]. The huge socioeconomic differences in the nine regions of Chongqing’s central city have led to inequalities in access to medical facilities, mainly including the level of nearby facilities, transportation networks, population density, and regional income levels [57]. At the horizontal level, the needs of the default population group are equal. The uneven distribution of urban and rural medical resources and public transportation’s “blank space” directly affect equitable accessibility. At the vertical level, differences in income levels and population density between regions indirectly affect population groups. Different population subgroups have different needs, some more than others. Theoretically, the greater the number of people in the region, the greater the demand potential of the population group, and the more adequate the allocation of medical resources should be to meet the conditions of health needs in the region. However, the results are different; Yubei District has the highest total population among the nine districts, but the proportion of medical facilities’ accessible service area is lower. In contrast, Yuzhong District and Beibei District have more medical resources in areas with lower populations. It is worth noting that both areas are old cities with high income levels. We further conclude that equity in access to healthcare facilities is greatly affected by socioeconomic differences.

Overall, the configuration of medical facilities in Chongqing is improving. However, there is still the problem of unequal distribution of regional medical resources. The following two suggestions are proposed:1.Ensure barrier-free access to community health facilities. Understand that having accessible medical facilities or public transportation is a condition for meeting health needs. Therefore, it is necessary to optimize the spatial layout of community-level hospitals, achieve full coverage of the service area of community-level hospitals, and improve public transportation lines so that residents can get essential medical and health protection quickly. At the same time, regional socioeconomic differences should be considered. The needs of population groups in the Yubei District, Jiangbei District, and other areas with a large population base must be understood. Especially in new urban areas and rural areas, it is necessary to support the priority construction of community health centers, to scientifically plan specific locations, and to establish a reasonable service radius.

This will save money and improve efficiency and the healthcare system of the entire city. Let ordinary diseases and chronic diseases stay at the grassroots level, and let severe patients be transferred to suitable hospitals for treatment.

2.Optimize the layout of general hospitals. When selecting the site of medical facilities, the urban spatial structure and population coordination distribution should be considered to avoid the distribution of medical facilities being too concentrated in the central metropolitan area or the old city, which not only hinders the process of urbanization but also is not conducive to the aggregation of the population to the urban area and reduces the development vitality of the new metropolitan area. At present, the general hospitals in Chongqing are concentrated in the urban center, while the service area ratios of Yubei District, Banan District, and Beibei District are small. In subsequent planning, according to the population distribution, the general hospital should be appropriately relocated to divert patients from the central urban area. At the same time, by strengthening the construction of tertiary grade A and below hospitals, people can get rid of excessive dependence on tertiary grade A hospitals.

In the future, work will focus on improving the rules of public transportation road network data and conduct comparative research on accessibility under different modes of transportation. At the same time, the availability of mass spatial–temporal data offers a significant opportunity to evaluate the characteristics of public facilities, including medical services, parklands, etc. In addition, this study has some shortcomings. First, owing to data limitations, this study only takes the districts of the main urban area as the research object, and the refinement of future research units is worth considering. Second, residents’ preferences and the needs of socially vulnerable groups are not considered, and results may deviate from residents’ actual healthcare service accessibility.

## Figures and Tables

**Figure 1 ijerph-19-16224-f001:**
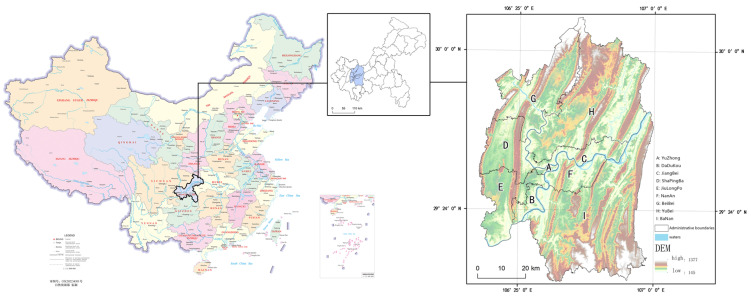
Map of Chongqing, China.

**Figure 2 ijerph-19-16224-f002:**
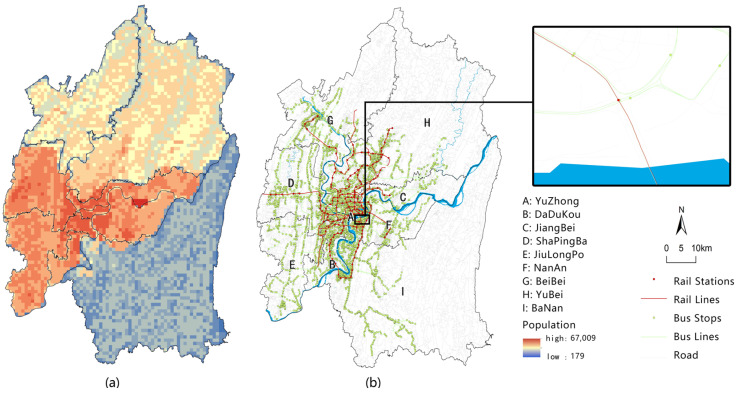
Population and traffic data of the main urban area: (**a**) population data of the main urban area of Chongqing; (**b**) public transport road network data in the main urban area of Chongqing.

**Figure 3 ijerph-19-16224-f003:**
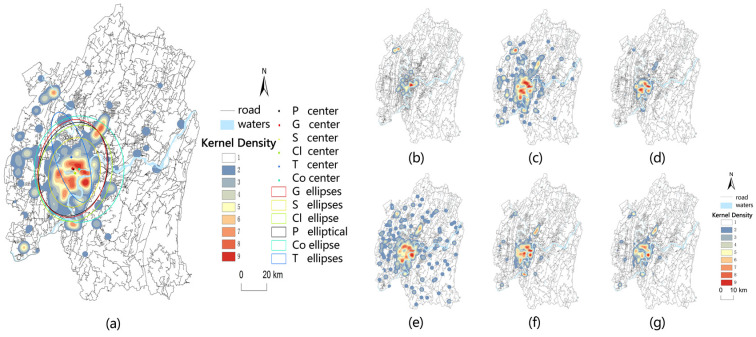
Kernel density analysis chart of various medical facilities: (**a**) general medical facilities; (**b**) tertiary grade A hospitals; (**c**) general hospitals; (**d**) specialist hospitals; (**e**) community-level hospitals; (**f**) clinics; (**g**) pharmacies.

**Figure 4 ijerph-19-16224-f004:**
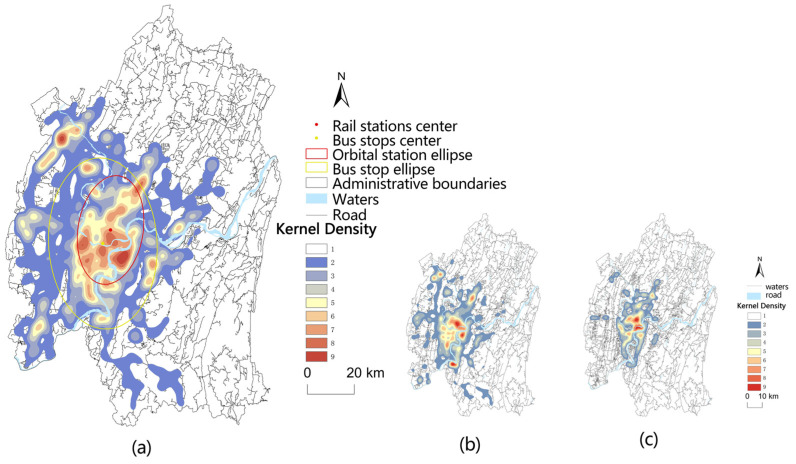
Kernel density analysis chart of public transport stations: (**a**) general public transport stations; (**b**) bus stops; (**c**) rail station.

**Figure 5 ijerph-19-16224-f005:**
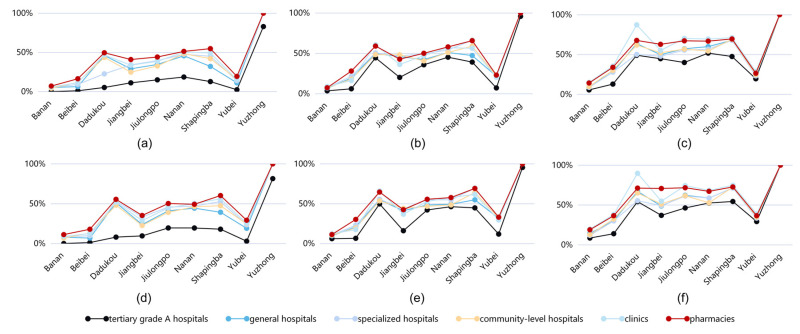
The average service population ratio and average service area ratio under different travel times of various medical facilities: (**a**–**c**) average service area ratio for 15 min, 30 min, and 60 min travel time; (**d**–**f**) average population ratio served at 15 min, 30 min, and 60 min travel time.

**Figure 6 ijerph-19-16224-f006:**
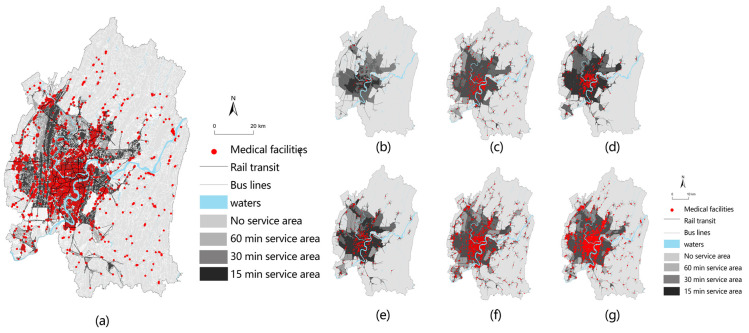
Service areas for various types of medical facilities: (**a**) general medical facilities; (**b**) tertiary grade A hospitals; (**c**) general hospitals; (**d**) specialist hospitals; (**e**) community-level hospitals; (**f**) clinics; (**g**) pharmacies.

**Figure 7 ijerph-19-16224-f007:**
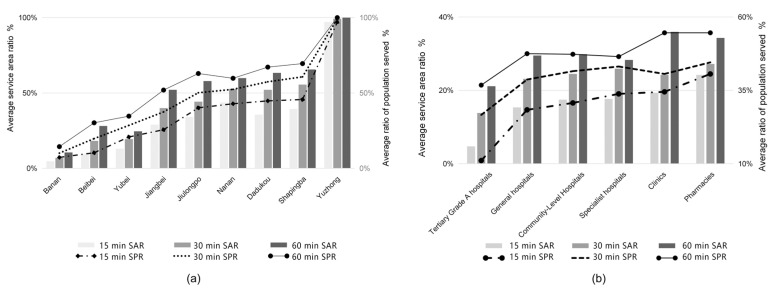
Accessibility of medical facilities: (**a**) average service area rate and average service population rate by the district; (**b**) the average service area rate and average service population rate of various types of medications.

**Table 1 ijerph-19-16224-t001:** Nearest Neighbor Ratio for all types of medical facilities.

Classify (Shorthand)	$\bar{r_{1}}$	$\bar{r_{\varepsilon }}$	R	Spatial Structure Type
tertiary grade A hospitals (T)	1741.02	7250.47	0.24	aggregation
general hospitals (G)	739.10	1795.44	0.41	aggregation
specialist hospitals (S)	330.15	1305.47	0.25	aggregation
community-level hospitals (CO)	1068.03	1772.59	0.60	aggregation
clinics (CL)	260.44	706.41	0.37	aggregation
pharmacies (P)	79.34	380.39	0.21	aggregation

**Table 2 ijerph-19-16224-t002:** Nearest Neighbor Ratio for public transport.

Classify	$\bar{r_{1}}$	$\bar{r_{\varepsilon }}$	R	Spatial Structure Type
Rail stations	1164.27	2860.85	0.41	agglomeration
Bus stops	291.69	606.15	0.48	agglomeration

## Data Availability

Not applicable.

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
