# Peer review of "Spatial Accessibility Analysis of Medical Facilities Based on Public Transportation Networks"

_ijerph, 2022, doi:10.3390/ijerph192316224_

Round 1

Reviewer 1 Report (Previous Reviewer 2)

Revisions have been made in the paper, however, largely focusing on why the research methods were selected (Page 2-3). Relevant maps were provided.

However, where is the conclusion section? According to the results of this study, more in-depth implications for other cities with similar topography should be discussed, in the context of global studies.

Author Response

Thank you for your suggestion.

Discussion, conclusions and future work are presented in Section 4. Among them, other cities with similar topography are discussed below. (Chapter Four, Paragraph Four)

Our findings suggest that the distribution of medical facilities is related to Chongqing's topography, transportation convenience, and population density. Extensive medical facilities are mainly located around major transportation arteries. Meanwhile, small clinics and pharmacies are mainly located in densely populated neighborhood streets. In contrast, inaccessible medical facilities are mainly located in the congested old urban areas and sparsely populated new urban areas. The problem of inaccessibility is affected by the distribution of medical facilities and public transport stations, which provides an idea for the location and layout of cities belonging to mountainous areas. The mountainous regions of the United States (Arizona, Colorado, New Mexico, Nevada, and Utah) use rail transit systems to help meet traffic demand, and distance from transit stations increases the decline in real estate values. Hong Kong, a representative of China's mountainous cities, found that public transport provided much faster travel times from all residential areas to the park. The average travel time of residents of crowded public housing exceeds that of residents of private housing. Therefore, vulnerable groups are more vulnerable to public transportation and medical facilities.

Reviewer 2 Report (Previous Reviewer 1)

The manuscript has improved a lot after the revision. However, there is still room for improvement if the authors want to attract a broader audience. My previous comments are being addressed in some way, but it is not easy to track down what has been revised according to the response letter. 

Author Response

Thanks for your comment. This revision adopts the "Track Changes" function to view any changes quickly. Attach a response to your previous comments (paragraph positioning instructions added).

Reviewer 3 Report (New Reviewer)

Dear authors,

Thank you for giving me the opportunity to review the manuscript titled “Spatial accessibility analysis of medical facilities based on public transportation networks”. This paper studies the spatial distribution of medical facilities and public transportation stations in the main urban area of Chongqing (China). 13,754 medical with spatial location data (including tertiary-grade A hospitals, General hospitals, specialist hospitals, Community-Level Hospitals, Clinics and pharmacies) have been obtained. It is a rigorous scientific contribution and achieves the proposed objectives. Both the Abstract and title accurately describe the contents of the manuscript. All elements of the manuscript relate logically to the to the study's statement of purpose.

However, some recommendations have suggested to help in areas of deficiency:

General comments:

·      In the introduction, a phrase that identifies the main scope of the paper should be exposed before the end of the page 3 of 15.

·   Figure 2b: the letters that represent the nine districts are not clearly visible within the map (page 5 of 15)

·  Subsection 2.2.3 Kernel Density: Kernel density” should not been highlighted in grey (page 6 of 15)

·     In the version that I have downloaded in pdf, Table 1 is repeated in pages 6 and 7 of 15.

·         There are words that begin with a capital letter after a comma. For example, in subsection 3.1 “Based on the near analysis, The spatial distribution of medical facilities” (page 6 of 15) and in Discussion “Moreover, The spatial distribution of medical facilities…” (page 11 of 15).

·     You should use capital letters in Discussion: 1. Focus and 3. Optimize.

·     Only references 30 and 33 have DOI numbers (Digital Object Identifier). Although DOI numbers are highly encouraged, they are not mandatory but for this journal. Use a consistent referencing.

Comments on content:

This paper is overall a study of spatial distribution of medical facilities and, therefore, it is an analysis of spatial inequalities. In this sense:

1.      It should be remarked that this paper contains implicitly two types of spatial inequalities that intersect:

·         Inequalities in the access to medical facilities  

·         Inequalities in the access to public transport

2.      One of the limitations of this work is that it only considers the supply side. This paper is an analysis of horizontal spatial inequity in the access to medical facilities. In this sense, it is implicitly hypothesized equal use across different population subgroups. However, considering only horizontal equity means, in fact, inequity. This occurs because similar time of access to medical facilities across different population groups does not guarantee equity. In fact, different population subgroups have different needs. Therefore, the analysis of fair accessibility to medical facilities must include vertical inequity (give different treatment to people with different needs; In other words, the more needs, the more resources) (Starfield, 2011). The ability to identify socially disadvantaged groups and locations is central to public policies that shape health systems. And this is crucial to removing any systematic spatial disparities and protecting vulnerable populations and locations.

There is a need to evaluate the horizontal and vertical dimensions in the assessment of equity in healthcare access. Therefore, socioeconomic differences among the nine analyzed districts characteristics are relevant to identify the double level of vulnerability of each district. First, the vulnerability due to the level of accessibility to medical facilities and to public transportation stations accessibility. Second, the vulnerability due to the socioeconomic differences inter- and intra-districts (there are locations and subgroups of population more affected by inaccessibility, both to public transport and medical facilities). In short, your paper should gain if you highlight the spatial socioeconomic differences in the analysed districts, specifying the level of vulnerability of the locations. It should be shown if there are substantial socioeconomic differences among the nine districts included in the paper, because it has been evidenced that inequalities in the access to medical facilities are affected by socioeconomic differences. This would be considered a plus for your scientific work.

3.      Related to the three suggestions that are proposed in page 12 of 15 in discussion section, “focus on the construction of community medical facilities” and “Accelerate the construction of community medical facilities in residential areas of new cities” will not guarantee reducing horizontal and vertical inequalities and meeting the accessibility needs of the entire population. It should be remarked that having accessible medical facilities or public transportation is a condition to meet health needs, but this should be combined with the analysis of vertical inequalities. Moreover, “construction” does not mean a fair provision of services for two main reasons: 1. Construction does not guarantee the provision of material and human resources. 2. Fair accessibility means considering vertical equity. These aspects should be incorporated into the discussion. It should also be remarked that accessibility plays a determining role in redressing social and spatial inequities, favoring or disadvantaging certain groups or locations. 

Congratulations for your work and insights reflected in the content of the manuscript. I hope my comments will be of value to you.

 Kind regards,

The Reviewer

Reference:

Starfield, B. (2011). The hidden inequity in health care. International journal for equity in health10(1), 1-3.

Author Response

Thank you for your suggestion. Your comments are of high value to us.

Round 2

Reviewer 3 Report (New Reviewer)

Dear authors,

The suggested changes have been incorporated. The manuscript has been revised in accordance with the reviewer comments. Therefore, it may be accepted in present form.

Kind regards,

The reviewer

This manuscript is a resubmission of an earlier submission. The following is a list of the peer review reports and author responses from that submission.

Round 1

Reviewer 1 Report

This paper presents a study that evaluates the spatial accessibility of medical facilities in the city of Chongqing. The study used the nearest analysis, kernel density, standard distance, and standard deviational ellipse method to analyze the spatial differentiation characteristics of medical facilities based on the public transportation networks. The research has its value; however, I don’t recommend publication in its current form due to the following reasons.

1.       The findings of the spatial distribution of medical facilities reflect the status quo in the city of Chongqing, which is highly dependent on its history and special terrain (famous as a mountain city). I don’t see how to integrate the knowledge into new spatial domains (other cities or other countries).

2.       The introduction of the paper doesn’t clearly outline the research gaps in the field. There is a lack of literature reviews in the field.

3.       The method used to measure spatial accessibility is not well justified.

4.       The geographic center is not what Standard Distance calculates. According to ArcGIS Help document, the Standard Distance measures the degree to which features are concentrated or dispersed around the geometric mean center, which generates a circle polygon around the mean center. The more proper method to calculate what you describe here is using the "Mean Center" tool in ArcGIS.

5.       It is unclear how the different types of medical service facilities are classified among tertiary grade A hospitals, General hospitals, specialist hospitals, Community-Level Hospitals, and Clinics.

6.       The network analysis considers walking and public transport (metro and bus). Is there a reason that driving is excluded from the analysis? As we know, for emergency cases, patients are transported to the nearest hospital by either private cars, taxis, or ambulances.

7.       Page 7: When measuring the Tertiary Grade A hospital accessibility, the conclusion assumes that residents of a district only access the hospital in that district. For the "Tertiary Grade A Hospital", it normally aims to serve a larger region but not only the residents of the district where the hospital is located.

8.       The manuscript is poorly organized. Also, extensive editing of the English language and style is required. There are many locations with citation errors (“Error! Reference source not found”). All the section numbers are “1”, “1.1”, or “1.1.1”.

9.       Figures and Tables are poorly presented. For Figure 1, What do the values of the Population/person mean? population density? what's the unit? Table 1 is unreadable.

Reviewer 2 Report

This paper examines the spatial accessibility of different levels of medical facilities in Chongqing, China. I have the following suggestions/ concerns.

(1) This study focused on the case of Chongqing City. Why? More solid reasons should be given to choose this study case.

Given the unique topography of Chongqing (i.e., mountainous region), specific discussions should be made, e.g., comparisons with studies of similar regions/ cities. Further, relevant introduction of Chongqing should be provided, including a map to indicate the location of Chongqing in China.

(2) Literature review of this paper is far from adequate, e.g., studies of other countries, relevant methods (most commonly used), studies of other cities in China, etc. It is unconvincing to employ time accessibility assessment with inadequate literature review.

(3) The title of this paper is “based on multiple transportation network”. However, this study largely examined the mode of walking, rail transit, and bus. The authors stated that “travel methods of urban residents have become more advocating the way of public transportation”. Are there any statistics to support this statement?

(4) Discussions are not closely related to the empirical results. What are the exact implications based on the empirical results? More in-depth discussions should be made. Comparisons should be made between results of this study and existing literature.

Current discussions are limited to the city of Chongqing. What are the implications to other cities, especially those with similar topography?

(5) The rationale to employ “The stop time was 30 s. The walking speed was 1.2 m/s, rail transit speed was 70 km/h, and the speed of buses was 40 km/h” should be discussed. Any statistics/ literature to support these speed data?

(6) The numbering of sessions need double-check and further revisions. Punctuations should also be double-checked. Some of them are Chinese punctuation marks.

Reviewer 3 Report

The subject of the paper is interesting, but after reading it carefully, the following issues were formulated.

- no "Lietarture review" section; there is a lack of a review of the world literature, ie from outside China, concerning the study of the transport accessibility of various objects and a review of the literature regarding research methods; there is also no overview of other similar case studies;

- the content of the paper is a case study, but there is no indication of what is new to learning from the presented calculation example;

- no "Conclusion" section;

- very poorly developed section on methodology, there are practically no new elements of this methodology;

- it seems that the text should be significantly improved in terms of its translation into English; for example: page 2: the text is incomprehensible: "(...) not considering the number of station routes, station stop times, etc."; What are the "station routes" if nothing has been written about rail transport before?

- the numbering of individual sections requires sorting;